# DG-GAN: THE GAN WITH THE DUALITY GAP

## ABSTRACT

Generative Adversarial Networks (GANs) are powerful framework for modeling complex and high dimensional data. The training of GANs is difficult because the optimization is a min-max problem. This paper understands GANs from the perspective of duality gap and shows that the duality gap can be a good metric to evolution the difference between the true data distribution and the distribution generated by generator. Training GANs using the duality gap can provide competitive results. Furthermore, we establish the generalization error bound of the duality gap to help design the neural network architecture and select the sample size.

## 1 INTRODUCTION

In the past few years, Generative Adversarial Networks (GANs) (Goodfellow et al., 2014) are impactful because it has shown lots of great results for many AI tasks, such as image generation, dialogue generation, and images inpainting (Abadi & G Andersen, 2016; Goodfellow, 2016; Ho & Ermon, 2016). Differing from other unsupervised learning methods for model generation that concentrate on the hard optimization of the measure of distribution fit such as the maximum likelihood method, GANs, which are a kind of methods of implicit models (Mohamed & Lakshminarayanan, 2017; Tran et al., 2017), can be seen as a game between two networks, the generator and the discriminator. Training GANs will improve the two networks' capability synchronously. Denote the discriminator as $f$ and a generator as $g$. The objective of GANs is

$$\inf_g \sup_f V(f,g) = \underset{x \sim p_{data}}{E}[\phi(f(x))] + \underset{x \sim p_z}{E}[\phi(1 - f(g(x)))], \tag{1}$$

where $p_{data}$ is the true data distribution and $p_z$ is the standard Gaussian distribution. Here, the goal of $f$ is to discriminate the difference between two distributions and the goal of $g$ is to generate a distribution with the Gaussian noise. Therefore the problem of GANs is a min-max problem. The minimization problem is to search for the optimal discriminator $f$ that can distinguish two distributions as much as possible and the maximization problem is to find the optimal generator $g$ such that the discriminator can not find the difference. So the GAN is just like a game between these two players. This is in general a challenging task to find the best solution because it may be not a concave-convex min-max optimization. This means that the objective, denoted by $V(f,g)$, may not be a convex function when fixing $f$ and not a concave function when fixing $g$.

The first major problem of GANs is how to measure the difference between the generated distribution and the true data distribution. It means that there is no an unanimous metric to represent the difference between the true data distribution and the generated distribution (Borji, 2018). Different metrics have achieved different performances on the different benchmark datasets, although many state-of-the-art models can show similar results (Lucic et al., 2017). It is also difficult to know whether the generated distribution is close to the true distribution, and this is often observed by human eyes. Another problem is the convergence of the training algorithm of GANs, especially the global convergence. It means that if the original generator and discriminator are random, it is difficult to confirm that the generator and discriminator can converge to the ideal conclusion by training with given data. So the existed algorithms should be heuristic or it can get a bad result even we train the neural networks with lots of datasets. Although it can be proved that the generator and discriminator can converge to the local Nash equilibrium under some strong assumptions (Martin et al., 2017), many GAN algorithms can not converge globally (Gemp & Mahadeven, 2019),

In this paper, our main contributions are:

- We propose a new metric of GANs and prove that the metric can be an upper bound of the traditional metrics.

- We establish a generalization error bound under the new metric and show that the empirical metric can be viewed as the loss function for GANs.

- We propose an new algorithm with the new metric which demonstrates better results than state-of-the-art algorithms.

The remainder of this paper is organized as follows. In Section 2, some related work are reviewed. Section 3 gives the new metric named duality gap that can be seen as an upper bound of traditional metrics. In Section 4, we establish a generalization error bound under the new metric and show that the empirical duality gap can be viewed as the loss function for GANs. Section 5 and 6 provide the new algorithm and some experimental results. Finally, we give our conclusions and future work.

## 2    RELATED WORK

The problem of the GANs' metric and convergence has been extensively explored over the past few decades, and a substantial amount of work has been proposed in the categories of convergence and new metric. The duality gap has ever been suggested by Grnarova et al. (2018). However, they only take the original GAN (Goodfellow et al., 2014) into consideration. Theis et al. (2015) has showed that even though the log-likelihood of the data can be seen as a loss function to train a generative model and thus can be seen as a metric of GANs, it has severe limitations because it may generate some low quality models with a high likelihood. Tolstikhin et al. (2017) proposed to use the probability mass of the real data "covered" by the model distribution as a metric. They used a kernel density estimation method to approximate the density of generated models' distribution and this metric is more interpretable than the likelihood, making it easier to assess the difference in performance of the algorithms. One of the most famous metric of GANs is the inception score (IS) (Salimans et al., 2016), which uses a pre-trained neural network (the Inception Net (Szegedy et al., 2016) trained on the ImageNet (Deng et al., 2009)) to capture the desirable properties of generated samples. It can measure the quality of the generated models and discriminability. There are some modifications of IS such as (Martin et al., 2017; Gurumurthy et al., 2017) and so on. Furthermore, Martin et al. (2017) proposed Frechet Inception Distance(FID) between two Gaussian distribution for evaluating the quality of these models. However, even though these kinds of metrics can get some good enough results on some samples, the Gaussian assumption is not always right and the FID can not work well with the non-labeled datasets.

There are some other research concentrating on the metric to estimate the generated distribution. Such as Maximum Mean Discrepancy (MMD) (Gretton et al., 2012), which measures the dissimilarity between two probability distributions using samples drawn independently from each other. However, the MMD method's computation complexity is the quadratic in the sample size, which is difficult to train. Arora & Zhang (2017) proposed to use the birthday paradox test to evaluate GANs, this test approximates the support size of a discrete distribution and can also be used to detect mode collapse in GANs. Generative Adversarial Metric(GAM) is proposed by Jiwoong Im et al. (2016), which means exchanging discriminators or generators of two GANs and then comparing the two GANs by engaging them in a battle against each other. Image Retrieval Performance (Wang et al., 2016) evaluates GANs with an image retrieval measure, the main idea of which is to examine the badly modeled images. There are some research that view the GANs as a zero-sum game.Grnarova et al. (2018) proposed the duality gap, but the paper only takes the log-likelihood into consideration. Balduzzi et al. (2018) introduced the Hamiltonian mechanics in the games and designed an algorithm that can converge to the Nash Equilibrium faster, and this method has showed some desirable results if applying in GANs. Oliehoek et al. (2017) studied GANs from the view of game theory and suggested an algorithm of training GANs to the Nash equilibrium. Grnarova et al. (2017) considered the Nash equilibrium for semi-shallow GAN architectures and other more complex architectures.

## 3    THE DUALITY GAP

In the section, we give the definition of the duality gap. Because the duality gap comes from game theory, we give some knowledge of game theory at first.

**Definition 3.1.** *(Game) A strategy game is a tuple $< \mathcal{P}, \{S_i\}_{i=1}^n, \{u_i\}_{i=1}^n >$, where $\mathcal{P} = \{p_1, ..., p_n\}$ is the players sets, $S_i$ is the set of pure strategies for player i and $u_i$ is i's payoff real-valued function defined on the pure strategy profiles's set: $S = S_1 \times ... \times S_n$*

The key of the game theory is the Nash Equilibrium, which is a strategy profile such that no player can change his payoff unilaterally.

**Definition 3.2.** *(Nash Equilibrium) A Nash Equilibrium is a strategy profile $< s_1, ..., s_i, ...s_n > \in S$ s.t. $\forall < s_1, ..., s_i', ..., s_n > \in S$, we have $u_i(s_1, ..., s_i, ..., s_n) \geq u_i(s_1, ..., s_i', ..., s_n)$ for any player i.*

In this paper, we only discuss GANs with only two players, the game mentioned below are two-players' game.

**Definition 3.3.** *(Zero-sum game) A zero-sum game is a game with the two payoff functions $u_1(s_1, s_2)$ and $u_2(s_1, s_2)$ s.t. $u_1(s_1, s_2) + u_2(s_1, s_2) = 0$ for any $(s_1, s_2) \in S$*

For a two-players' zero-sum game, its equilibria also is called saddle point, which has some important properties and has attracted lots of attentions. Because the saddle point is difficult to research, this leads the difficulty of the GANs' research. About the equilibria, we have the following theorem:

**Theorem 3.1.** *In a zero-sum game, we have*

$$\sup_{s_2} \inf_{s_1} u_i(s_1, s_2) = \inf_{s_1} \sup_{s_2} u_i(s_1, s_2) = v \tag{2}$$

*where the v is called the value of the zero-sum game.*

The strategy $(s_1, s_2) \in S$ is called the maximin strategy. For these two players, they have different maximin strategies. The player 1's maximin strategy is $\widehat{s_1}$ such that $\sup_{s_2} u_i(\widehat{s_1}, s_2) = v$ and the player 2's maximin strategy is $\widehat{s_2}$ such that $\inf_{s_1} u(s_1, \widehat{s_2}) = v$. Furthermore, if we combine the two maximin strategies of these two players, we can achieve an equilibrium.

### 3.1 THE DUALITY GAP OF GANS

The traditional machine learning problem can be seen as an optimization problem. The objective to be minimized is denoted by a loss function. However, because the GAN objective is a min-max problem, it can be seen as the zero-sum game, with the 2 players being the generator and the discriminator. We will introduce the duality gap metric, which can be used to estimate the ability of the generators and the discriminators, and the relationship of duality gap and the classical metric–$\mathcal{F} - distance$ when the generator's and discriminator's capacity are unbounded.

A zero-sum game comes from game theory, consisting of 2 players $D$(Discriminator) and $G$(Generator) with their strategy-fields $\mathcal{F}$ and $\mathcal{G}$. A function $V : \mathcal{F} \times \mathcal{G} \to R$ is the utilities of the 2 players. By selecting $(f, g) \in \mathcal{F} \times \mathcal{G}$, the $D$'s utility is $+V$ and the $G$'s utility is $-V$. The goal of the 2 players is to maximize the worst case utility, which is

$$\sup_{f \in \mathcal{F}} \inf_{g \in \mathcal{G}} V(f, g) \quad \& \quad \inf_{g \in \mathcal{G}} \sup_{f \in \mathcal{F}} V(f, g). \tag{3}$$

The strategy $(f^*, g^*) \in \mathcal{F} \times \mathcal{G}$ is called (Pure) Equilibrium if it satisfies that

$$\sup_{f \in \mathcal{F}} V(f, g^*) = \inf_{g \in \mathcal{G}} V(f^*, g). \tag{4}$$

According to the above discussion, the GANs' duality gap metric of the pure strategy can be defined.

**Definition 3.4.** *(Duality Gap of GANs) Given 2 strategy fields $\mathcal{F}$ and $\mathcal{G}$, strategy $(f^*, g^*) \in \mathcal{F} \times \mathcal{G}$, a convex function $\phi$, a true data distribution $p_{data}$, and a Gaussian distribution $p_z$, the duality gap of $(f^*, g^*)$ is*

$$DG(f^*, g^*) := \sup_{f \in \mathcal{F}} V(f, g^*) - \inf_{g \in \mathcal{G}} V(f^*, g) \tag{5}$$

*Here the $V(f, g)$ is the the function that GANs concentrate on:*

$$V(f, g) = \mathop{E}_{x \sim p_{data}} [\phi(f(x))] + \mathop{E}_{x \sim p_z} [\phi(1 - f(g(x)))] \tag{6}$$

## 3.2 DUALITY GAP AS A METRIC

The traditional metric used in GANs is a kind of distance between two distribution, denoted by $\mathcal{F} - distance$.

**Definition 3.5.** *($\mathcal{F} - distance$) Given a function space $\mathcal{F} = \{f : R^d \to R | f \in \mathcal{F} \Leftrightarrow 1 - f \in \mathcal{F}\}$. A convex function $\phi$, a distribution $p_{data}$, a Gaussian distribution $p_z$ and a generator $g$, then*

$$d_{\mathcal{F},\phi}(p_{data}, p_g) = \sup_{f \in \mathcal{F}} \underset{x \sim p_{data}}{E} [\phi(f(x))] + \underset{x \sim p_g}{E} [\phi(1 - f(x))] - 2\phi(\frac{1}{2}). \tag{7}$$

*So the $\mathcal{F} - distance$ can be written as*

$$d_{\mathcal{F},\phi}(p_{data}, p_g) = \sup_{f \in \mathcal{F}} V(f, g), \tag{8}$$

*where $V(f, g)$ has been defined in equation (4).*

**Remark 3.1.** *$\mathcal{F} - distance$ is a distance between two distributions: $p_{data}$ and $p_g$. For a special case when $\phi(x) = x$ and $\mathcal{F} = \{f : R^d \to R | L_f < \infty\}$, then the $\mathcal{F} - distance$ is Wasserstein-Distance, where $L_f$ is the Lipschitz constant of $f$.*

The next theorem shows that the duality gap can be an upper bound of $\mathcal{F} - distance$ with the given condition.

**Theorem 3.2.** *If for any distribution $p$, $\exists g \in \mathcal{G}$, s.t. $g(z) \sim p$ where $z \sim p_z$ that is a given Gaussian distribution. Assuming $\{f : R^d \to R | L_f < \infty\} \subset \mathcal{F}$, then*

$$\sup_{f \in \mathcal{F}} V(f, g^*) - \inf_{g \in \mathcal{G}} V(f^*, g) \geq \sup_{f \in \mathcal{F}} V(f, g^*) - 2\phi(\frac{1}{2}) \geq 0 \tag{9}$$

*Proof.* Observe that

$$\sup_{f \in \mathcal{F}} V(f, g^*) - \inf_{g \in \mathcal{G}} V(f^*, g) \geq \sup_{f \in \mathcal{F}} V(f, g^*) - 2\phi(\frac{1}{2}) \Leftrightarrow \inf_{g \in \mathcal{G}} V(f^*, g) \leq 2\phi(\frac{1}{2}). \tag{10}$$

According the property of $\mathcal{G}$,

$$\inf_{g \in \mathcal{G}} V(f^*, g) \leq V(f^*, g)|_{p_g = p_{data}} = \underset{x \sim p_{data}}{E} [\phi(f^*(x)) + \phi(1 - f^*(x))] \leq 2\phi(\frac{1}{2}), \tag{11}$$

where the second inequality comes from the property of $\mathcal{F}$. Hence,

$$\sup_{f \in \mathcal{F}} V(f, g^*) - 2\phi(\frac{1}{2}) \geq V(f, g^*)|_{f = \frac{1}{2}} - 2\phi(\frac{1}{2}) = 0. \tag{12}$$

$\square$

The theorem above shows that if the discriminator and generator have unbounded capacities, the $\mathcal{F} - distance$ can be a metric to discriminate the $p_g$ and $p_{data}$ and the duality gap is an upper bound of the $\mathcal{F} - distance$.

## 4 THE GENERALIZATION ERROR BOUND ON THE DUALITY GAP

Considering the training of GANs with the new metric, we first establish the generalization error bound of the duality gap. The generalization error bound is the gap between the training error and the test error. In general, the gap can be replaced by the empirical error and the population error when assuming the test datasets are infinite. The generalization error bound in general depends the sample size and the complexities of the function spaces of the discriminators and generators. So establishing the generalization error bound can guide the design of these two neural networks and select the sample size. The generalization error bound for vanilla GANs has been studied in the literature. For example, spectral weight normalization (Miyato et al., 2018) is used to establish a tight bound for GANs by (Jiang et al., 2019).

The generalization error bound of the unsupervised learning is always related to the complexity of the function space. We use Rademacher complexity to characterize the capacity of the function space. Because GANs have two function spaces $\mathcal{F}$ and $\mathcal{G}$ and the duality gap is related to these two spaces, the complexities of $\mathcal{F}$ and $\mathcal{G}$ are the keys to establish the duality gap's generalization bound.

**Definition 4.1.** *(Rademacher Complexity) Given a function space $\mathcal{F}$ and a random sample $X = \{x_1, ..., x_n\}$ where $x_i \sim \mu$, then the empirical and the expected Rademacher Complexity are, respectively,*

$$\widehat{\mathcal{R}}_X(\mathcal{F}) = \underset{\epsilon}{E}[\underset{f \in \mathcal{F}}{sup} \frac{1}{n} \sum_{i=1}^{n} \epsilon_i f(x_i)], \qquad \widehat{\mathcal{R}}_{n,\mu}(\mathcal{F}) = \underset{X \sim \mu^n}{E}[\widehat{\mathcal{R}}_X(\mathcal{F})], \qquad (13)$$

*where the distribution of $\epsilon = (\epsilon_1, ..., \epsilon_n)$ satisfies that $P(\epsilon_i = 1) = P(\epsilon_i = -1) = \frac{1}{2}$.*

The generalization error bound of the duality gap concentrates on the gap between the population duality gap denoted by $DG$ and the empirical duality gap denoted by $\widehat{DG}$,

$$\widehat{DG}(f^*, g^*) = \underset{f \in \mathcal{F}}{sup} \widehat{V}(f, g^*) - \underset{g \in \mathcal{G}}{inf} \widehat{V}(f^*, g), \qquad (14)$$

where

$$\widehat{V}(f,g) = \underset{x \sim \widehat{p}_{data}}{E}[\phi(f(x))] + \underset{z \sim \widehat{p}_z}{E}[\phi(1 - f(g(z)))] = \sum_{i=1}^{n} \frac{\phi(f(x_i))}{n} + \sum_{i=1}^{m} \frac{\phi(1 - f(g(z_i)))}{m}, \quad (15)$$

and the $x_i$ are selected from observed data and the $z_i$ are sampled from a standard Gaussian distribution.

**Theorem 4.1.** *If the true data sample $X$ and the Gaussian-distribution sample $Z$ are bounded and the bound is denoted by $B_X$ and $B_Z$, and the $\exists L_\mathcal{F}, L_\mathcal{G}$ s.t. $\forall f \in \mathcal{F}$ and $g \in \mathcal{G}$, the Lipschitz constant of $f$ is less than $L_\mathcal{F}$, and the Lipschitz constant of $g$ is less than $L_\mathcal{G}$. Then with probability at least $1 - 3\delta$*

$$|DG - \widehat{DG}| \leq 4\rho_\phi \widehat{\mathcal{R}}_X(\mathcal{F}) + 2\rho_\phi L_\mathcal{G} \widehat{\mathcal{R}}_{g*(Z)}(\mathcal{F}) + 2\rho_\phi L_\mathcal{F} \widehat{\mathcal{R}}_Z(\mathcal{G})$$
$$+ 12\rho_\phi L_\mathcal{F} B_X \sqrt{\frac{log\frac{2}{\delta}}{2n}} + 12\rho_\phi L_\mathcal{F} L_\mathcal{G} B_Z \sqrt{\frac{log\frac{2}{\delta}}{2m}}. \qquad (16)$$

For GANs, the two players generator and discriminator are approximated by deep neural networks, so the Rademacher Complexity is a function of the two neural networks' parameter. Supposing $f \in \mathcal{F}$ and $g \in \mathcal{G}$, then the $f$ and $g$ can be written as the form of a composition of a sequence of function, i.e.,

$$f = a_H(M_H(a_{H-1}(M_{H-1}(...a_1(M_1(\cdot))...)))),$$
$$g = b_{H'}(N_{H'}(b_{H'-1}(N_{H'-1}(...b_1(N_1(\cdot))...)))), \qquad (17)$$

where $a_i$ and $b_i$ are activation functions, $M_i$ and $N_i$ are matrices. Assume that the Lipschitz constants of $a_i$ and $b_i$ are less than 1. This is true for many popular activation functions such as ReLU. We also assume $||M_i|| \leq B_i$ and $||N_i|| \leq B_i'$. Let $d_f$ and $d_g$ denote the widths of these two networks respectively.

**Lemma 4.1.** *For the empirical Rademacher Complexity given above,*

$$\widehat{\mathcal{R}}_X(\mathcal{F}) \leq \frac{4}{n} + \frac{12 B_X \prod_{i=1}^{H} B_i \sqrt{d_f^2 H log(2\sqrt{d_f n} H B_X \prod_{i=1}^{H} B_i)}}{\sqrt{n}}$$

$$\widehat{\mathcal{R}}_Z(\mathcal{G}) \leq \frac{4}{m} + \frac{12 B_Z \prod_{i=1}^{H'} B_i' \sqrt{d_g^2 H' log(2\sqrt{d_g m} H' B_Z \prod_{i=1}^{H'} B_i')}}{\sqrt{m}} \qquad (18)$$

$$\widehat{\mathcal{R}}_{g*(Z)}(\mathcal{F}) \leq \frac{4}{m} + \frac{12 B_Z \prod_{i=1}^{H} B_i \sqrt{d_f^2 H log(2\sqrt{d_f m} H B_{g*(Z)} \prod_{i=1}^{H} B_i)}}{\sqrt{m}}$$

This above theorem shows that the empirical Rademacher Complexity's bound depends on these two neural networks' architectures, especially the width and the depth. When training GANs, we generate a noise for every iteration, so we can claim that $m \gg n$. Combining these two theorems, we obtain

**Theorem 4.2.**

$$|DG - \widehat{DG}| \leq \frac{48\rho_\phi B_Z \prod_{i=1}^{H} B_i \sqrt{d_f^2 H log(2\sqrt{d_f n} H B_Z \prod_{i=1}^{H} B_i)}}{\sqrt{n}} + 12\rho_\phi L_{\mathcal{F}} B_X \sqrt{\frac{log\frac{2}{\delta}}{2n}} + o(n^{-\frac{1}{2}}).$$

(19)

Based on (19), if the empirical duality gap $\widehat{DG}(f^*, g^*) \leq \epsilon$, we can establish the population bound of the $\mathcal{F} - distance$ such that

$$
\begin{aligned}
d_{\mathcal{F},\phi}(p_{data}, p_{g^*}) \leq & DG(f^*, g^*) \\
\leq & |DG(f^*, g^*) - \widehat{DG}(f^*, g^*)| + \widehat{DG}(f^*, g^*) \\
\leq & \frac{48\rho_\phi B_Z \prod_{i=1}^{H} B_i \sqrt{d_f^2 H log(2\sqrt{d_f n} H B_Z \prod_{i=1}^{H} B_i)}}{\sqrt{n}} \\
& + 12\rho_\phi L_{\mathcal{F}} B_X \sqrt{\frac{log\frac{2}{\delta}}{2n}} + o(n^{-\frac{1}{2}}) + \epsilon.
\end{aligned}
$$

(20)

## 5 THE ALGORITHM

According to the Sections 3 and 4, we know that the population duality gap is an upper bound of $\mathcal{F}$-distance and the gap between population duality gap and empirical duality gap can be arbitrarily small. Our theories imply that the empirical duality gap can be used as a loss function for training GANs. Note that many classical algorithms use $\mathcal{F}$-distance as the loss function. We develop a new algorithm using duality gap as the loss function. We focus on WGAN-GP, the loss function of which is

$$d_{\mathcal{F},\phi}(p_{data}, p_{g^*}) + \lambda \mathop{E}_{\hat{x} \sim P_{\hat{x}}} \left[ (\|\nabla_{\hat{x}} f(\hat{x})\|_2 - 1)^2 \right]. \tag{21}$$

Instead, our loss function is written as

$$DG(f^*, g^*) + \lambda \mathop{E}_{\hat{x} \sim P_{\hat{x}}} \left[ (\|\nabla_{\hat{x}} f(\hat{x})\|_2 - 1)^2 \right] \tag{22}$$

where $\hat{x} = \epsilon x + (1 - \epsilon)\tilde{x}$, $\epsilon \sim U(0, 1)$, $x \sim p_{data}$, $\tilde{x} = G(z)$, $z \sim p_z$. The details of the algorithm is given in Algorithm 1. We call our method DG-GAN, the GAN with the duality gap.

## 6 NUMERICAL EXPERIMENTS

In order to test our method, We conduct experiments using the duality gap on some datasets such as a toy dataset, MNIST, CIFAR-10, qnd so on. Then we compare our method DG-GAN with classic GAN models such as WGAN and WGAN-GP. The experiment results show that there are significant practical benefits to using our method over the traditional methods. There are two main benefits: (1) DG-GAN provides a good metric suggesting the generator's convergence and sample's quality. (2) Our method using duality gap as loss function has faster rate of convergence.

We train DG-GANs on CIFAR-10, and compare our method with WGANs. Specifically, we adopt a 4-layer CNN as the generator and a 3-layer CNN as the discriminator. In the following, $\lambda$ is 10. Number of discriminator iterators per generator iterators is 5. We run 20K iterations in all the experiments on CIFAR-10. Figure 1 shows the Wasserstein Distance on CIFAR-10 datasets training with algorithm 1, And for quantitative assessment of our generated examples, we use the inception score (Salimans et al. (2016)). Figure 2 shows the Inception score on CIFAR-10 datasets and Figure 3 shows the image generated after 20K iterations by the generator on CIFAR-10.

In addition to the inception scores of the two methods, we also calculate the FID (Fréchet Inception Distance) of them. For WGAN-GP, after 20K iteration's training, the FID between generated distribution and true distribution is 54.4, however for DG-GAN, it is 45.6. These observations, based on IS and FID, show that DG-GAN can provide a better quality of generated samples.

---

**Algorithm 1:** Learning parameters for BPR

---

   **input** :

         sample real data $x \sim P_{data}$;

         latent variable $z \sim P_z$;

         a random number $\epsilon \sim U[0,1]$;

   **output:**

         Generator parameter $\theta$;

1   initialize the generator parameter $\theta$ and the discriminator parameter $\omega$ and Adam parameter$\alpha = 0.0001, \beta_1 = 0, \beta_2 = 0.9$;

2   **while** $\theta$ *not convergence* **do**

3      $\omega^* = \omega$;

4      **for** $t = 1, ..., n_{critic}$ **do**

5         **for** $t = 1, ..., m$ **do**

6           $\tilde{x} \leftarrow g_\theta(z) \; \hat{x} \leftarrow \epsilon x + (1 - \epsilon) \tilde{x}$

             $L^{(i)} \leftarrow f_\omega(\tilde{x}) - f_\omega(x) + \lambda \left( \left( ||\nabla_{\hat{x}} f_\omega(\hat{x})||_2 - 1 \right)^2 \right.$

7         **end**

8         $\omega \leftarrow Adam \left( \nabla_\omega \frac{1}{m} \sum_{i=1}^m L^{(i)}, \omega, \alpha, \beta_1, \beta_2 \right)$

9      **end**

10     Sample a batch of latent variables $\left\{ z^{(i)} \right\}_{i=1}^m \sim p_z$

       $\theta \leftarrow Adam \left( \nabla_\theta \frac{1}{m} \sum_{i=1}^m -f_\omega(g_\theta(z)), \theta, \alpha, \beta_1, \beta_2 \right)$;

11     $\omega = \omega^*$;

12     Sample a batch of latent variables $\left\{ z^{(i)} \right\}_{i=1}^m \sim p_z$

       $\theta \leftarrow Adam \left( \nabla_\theta \frac{1}{m} \sum_{i=1}^m -f_\omega(g_\theta(z)), \theta, \alpha, \beta_1, \beta_2 \right)$;

13     **for** $t = 1, ..., n_{critic}$ **do**

14       **for** $t = 1, ..., m$ **do**

15          $L^{(i)} \leftarrow f_\omega(\tilde{x}) - f_\omega(x) + \lambda \left( \left( ||\nabla_{\hat{x}} f_\omega(\hat{x})||_2 - 1 \right)^2 \right.$

16       **end**

17       $\omega \leftarrow Adam \left( \nabla_\omega \frac{1}{m} \sum_{i=1}^m L^{(i)}, \omega, \alpha, \beta_1, \beta_2 \right)$

18     **end**

19   **end**

---

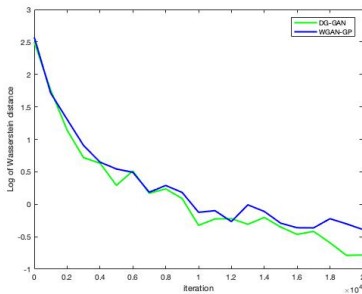

Figure 1: Wasserstein Distance on CIFAR-10

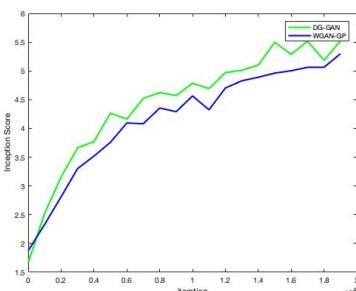

Figure 2: Inception Score on CIFAR-10

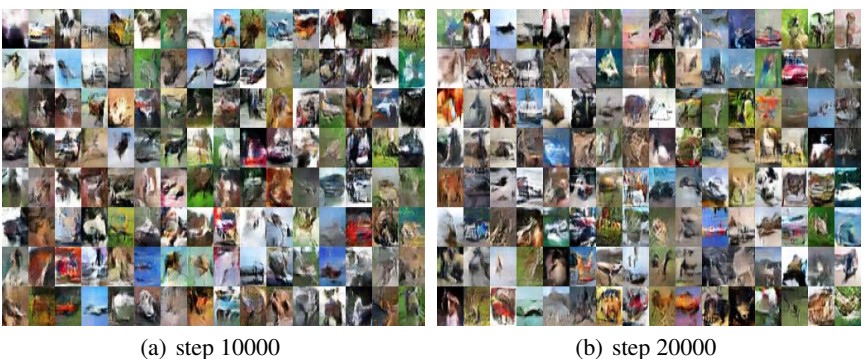

(a) step 10000           (b) step 20000

Figure 3: result on CIFAR-10

# 7 CONCLUSION

In this paper, we introduce a new metric for GANs, which can bound the traditional metric under several assumptions. We establish the generalization error bound of the new metric to help design the neural networks and select the sample size. We call this new framework DG-GAN. We compare the performance between DG-GANs and other classical GANs on benchmark datasets and DG-GAN has demonstrated competitive performance.

There are several future research directions. The first is to extend DG-GANs to autoencoder GANs, where we have an addition encoder network to learn the meaningful encoding. The second is to develop a formal hypothesis testing procedure to test whether the generated sample and the observed sample have the same distribution.

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

## A  THE PROOF OF THEOREMS

### A.1  THE PROOF OF THEOREM 4.1

The theorem 4.1 gives the generalization error bound of the duality gap with the Rademacher Complexity.

**proof.**

$$
| \sup_{f \in \mathcal{F}} V(f, g^*) - \inf_{g \in \mathcal{G}} V(f^*, g) - (\sup_{f \in \mathcal{F}} \widehat{V}(f, g^*) - \inf_{g \in \mathcal{G}} \widehat{V}(f^*, g))|
$$

$$
\leq | \sup_{f \in \mathcal{F}} V(f, g^*) - \sup_{f \in \mathcal{F}} \widehat{V}(f, g^*)| + | \inf_{g \in \mathcal{G}} \widehat{V}(f^*, g) - \inf_{g \in \mathcal{G}} V(f^*, g)|
$$

$$
\leq 2( \sup_{f \in \mathcal{F}} \underset{x \sim p_{data}}{E} [\phi(f(x))] - \underset{x \sim \widehat{p}_{data}}{E} [\phi(f(x))])
$$

$$
+ ( \sup_{f \in \mathcal{F}} \underset{x \sim p_z}{E} [\phi(1 - f(g^*(x)))] - \underset{x \sim \widehat{p}_z}{E} [\phi(1 - f(g^*(x)))])
$$

$$
+ ( \sup_{g \in \mathcal{G}} \underset{x \sim p_z}{E} [\phi(1 - f^*(g(x)))] - \underset{x \sim \widehat{p}_z}{E} [\phi(1 - f^*((x)))]) \tag{23}
$$

*Let $X = \{x_1, x_2, ..., x_i, ..., x_n\}$, $X' = \{x_1, x_2, ..., x_i', ...x_n\}$ and $\rho_\phi = ||\phi||_{Lip}$*

$$
| \sup_{f \in \mathcal{F}} \underset{x \sim p_{data}}{E} [\phi(f(x))] - \underset{x \sim \widehat{p}_{data}}{E} [\phi(f(x))] -
$$

$$
- \sup_{f \in \mathcal{F}} \underset{x \sim p_{data}}{E} [\phi(f(x))] - \underset{x \sim \widehat{p}'_{data}}{E} [\phi(f(x))]|
$$

$$
\leq \frac{1}{n} \sup_{f \in \mathcal{F}} |\phi(f(x_i)) - \phi(f(x_i'))| \leq 2 \frac{\rho_\phi}{n} L_{\mathcal{F}} B_X \tag{24}
$$

*Using McDiarmid's inequality, with probability at least $1 - \frac{\delta}{2}$*

$$
\sup_{f \in \mathcal{F}} \underset{x \sim p_{data}}{E} [\phi(f(x))] - \underset{x \sim \widehat{p}_{data}}{E} [\phi(f(x))]
$$

$$
\leq \underset{\widehat{p}_{data}}{E} [\sup_{f \in \mathcal{F}} \underset{x \sim p_{data}}{E} [\phi(f(x))] - \underset{x \sim \widehat{p}_{data}}{E} [\phi(f(x))]] + 2\rho_\phi L_{\mathcal{F}} B_X \sqrt{\frac{log\frac{2}{\delta}}{2n}} \tag{25}
$$

*And use McDiarmid's inequality again, with probability at least $1 - \frac{\delta}{2}$*

$$
E[\sup_{f \in \mathcal{F}} \underset{x \sim p_{data}}{E} [\phi(f(x))] - \underset{x \sim \widehat{p}_{data}}{E} [\phi(f(x))]]
$$

$$
\leq 2 \underset{x_i \sim p_{data}, \epsilon}{E} [\frac{1}{n} \sup_{f \in \mathcal{F}} \sum_{i=1}^{n} \epsilon_i \phi(f(x_i))]
$$

$$
\leq 2 \underset{\epsilon}{E} [\frac{1}{n} \sup_{f \in \mathcal{F}} \sum_{i=1}^{n} \epsilon_i \phi(f(x_i))] + 2\rho_\phi \sup_{f, x_i, x_i'} |f(x_i) - f(x_i')| \sqrt{\frac{\log \frac{2}{\delta}}{2n}} \quad (26)
$$

$$
\leq 2\rho_\phi \underset{\epsilon}{E} [\frac{1}{n} \sup_{f \in \mathcal{F}} \sum_{i=1}^{n} \epsilon_i f(x_i)] + 2\rho_\phi \sup_{f, x_i, x_i'} |f(x_i) - f(x_i')| \sqrt{\frac{\log \frac{2}{\delta}}{2n}}
$$

$$
= 2\rho_\phi \widehat{\mathcal{R}}_X(\mathcal{F}) + 4\rho_\phi L_\mathcal{F} B_X \sqrt{\frac{\log \frac{2}{\delta}}{2n}}
$$

*Here $\epsilon = (\epsilon_1, \epsilon_2, ..., \epsilon_n)$ and $P(\epsilon_i = 1) = P(\epsilon_i = -1) = 0.5$*
*So with probability at least $1 - \delta$*

$$
\sup_{f \in \mathcal{F}} \underset{x \sim p_{data}}{E} [\phi(f(x))] - \underset{x \sim \widehat{p}_{data}}{E} [\phi(f(x))]
$$

$$
\leq 2\rho_\phi \widehat{\mathcal{R}}_X(\mathcal{F}) + 6\rho_\phi L_\mathcal{F} B_X \sqrt{\frac{\log \frac{2}{\delta}}{2n}} \quad (27)
$$

*Similarly, with probability at least $1 - \delta$*

$$
\sup_{g \in \mathcal{G}} \underset{x \sim p_z}{E} [\phi(1 - f^*(g(x)))] - \underset{x \sim \widehat{p}_z}{E} [\phi(1 - f^*(g(x)))]
$$

$$
\leq 2\rho_\phi \cdot L_\mathcal{F} \widehat{\mathcal{R}}_Z(\mathcal{G}) + 6\rho_\phi \cdot L_\mathcal{F} L_\mathcal{G} B_Z \sqrt{\frac{\log \frac{2}{\delta}}{2m}}
$$

$$
\sup_{f \in \mathcal{F}} \underset{x \sim p_z}{E} [\phi(1 - f(g^*(x)))] - \underset{x \sim \widehat{p}_z}{E} [\phi(1 - f(g^*(x)))] \quad (28)
$$

$$
\leq 2\rho_\phi \cdot L_\mathcal{G} \widehat{\mathcal{R}}_Z(\mathcal{F}) + 6\rho_\phi L_\mathcal{F} L_\mathcal{G} B_Z \sqrt{\frac{\log \frac{2}{\delta}}{2m}}
$$

*So, we get the next inequality with probability at least $1 - 3\delta$*

$$
|DG - \widehat{DG}| \leq 4\rho_\phi \widehat{\mathcal{R}}_X(\mathcal{F}) + 2\rho_\phi \cdot L_\mathcal{G} \widehat{\mathcal{R}}_Z(\mathcal{F}) + 2\rho_\phi \cdot L_\mathcal{F} \widehat{\mathcal{R}}_Z(\mathcal{G})
$$

$$
+ 12\rho_\phi \cdot L_\mathcal{F} B_X \sqrt{\frac{\log \frac{2}{\delta}}{2n}} + 12\rho_\phi \cdot L_\mathcal{F} \cdot L_\mathcal{G} B_Z \sqrt{\frac{\log \frac{2}{\delta}}{2m}} \quad (29)
$$

## A.2 THE PROOF OF LEMMA 4.1

The lemma 4.1 gives the bound of the Rademacher Complexity

**proof.**

$$\|f(x) - f'(x)\|_\infty$$
$$\leq \|a_H(M_H(a_{H-1}(M_{H-1}(...a_1(M_1(x))...)))) - a_H(M'_H(a_{H-1}(M'_{H-1}(...a_1(M'_1(x))...))))\|_2$$
$$\leq \|a_H(M_H(a_{H-1}(M_{H-1}(...a_1(M_1(x))...)))) - a_L(M'_H(a_{H-1}(M_{H-1}(...a_1(M_1(x))...))))\|_2$$
$$+ \|a_H(M'_H(a_{H-1}(M_{H-1}(...a_1(M_1(x))...)))) - a_H(M'_H(a_{H-1}(M'_{H-1}(...a_1(M'_1(x))...))))\|_2$$
$$\leq \|M_H - M'_H\|_2 B_X \prod_{i=1}^{H-1} \|M_i\|_2 + \|M'_H\|_2 \|a_{H-1}(...a_1(M_1(x))...) - a_{L-1}(...a_1(M'_1(x))...)\|_2$$
$$\leq \cdots$$
$$\leq \sum_{i=1}^{H} B_X \prod_{j=1,j\neq i}^{H} \|M_j\|_2 \|M_i - M'_i\|_2 \leq \sum_{i=1}^{H} B_X \cdot \prod_{j=1,j\neq i}^{H} B_i \|M_i - M'_i\|_2$$

(30)

*For $\mathcal{M} = \{M \in R^{m\times n} : \|M\|_2 \leq B_i\}$, its' covering number $\mathcal{N}(\mathcal{M}, \epsilon, \|\cdot\|_2)$ satisfy*

$$\mathcal{N}(\mathcal{M}, \epsilon, \|\cdot\|_2) \leq (1 + \frac{min(\sqrt{m}, \sqrt{n})B_i}{\epsilon})^{mn}$$

(31)

*Hence,*

$$\mathcal{N}(\mathcal{F}, \epsilon, \|\cdot\|_\infty) \leq \prod_{i=1}^{H} \mathcal{N}(M_i, \frac{\epsilon}{LB_X \prod_{j=1,j\neq i}^{H} B_j}, \|\cdot\|_2)$$

(32)

*So, we have*

$$\mathcal{N}(\mathcal{F}, \epsilon, \|\cdot\|_\infty) \leq (1 + \frac{\sqrt{d_f}HB_X \prod_{i=1}^{H} B_i}{\epsilon})^{d_f^2 H}$$

(33)

*According to the relationship between Rademacher Complexity and covering number, we get*

$$\widehat{\mathcal{R}}_X(\mathcal{F}) \leq \frac{4}{n} + \frac{12B_X \prod_{i=1}^{H} B_i \sqrt{d_f^2 H log(2\sqrt{d_f n}HB_X \prod_{i=1}^{H} B_i)}}{\sqrt{n}}$$

(34)

*Similarly,*

$$\widehat{\mathcal{R}}_Z(\mathcal{G}) \leq \frac{4}{m} + \frac{12B_Z \prod_{i=1}^{H'} B'_i \sqrt{d_g^2 H' log(2\sqrt{d_g m}H'B_Z \prod_{i=1}^{H'} B'_i)}}{\sqrt{m}}$$
$$\widehat{\mathcal{R}}_{g^*(Z)}(\mathcal{F}) \leq \frac{4}{m} + \frac{12B_Z \prod_{i=1}^{H} B_i \sqrt{d_f^2 H log(2\sqrt{d_f m}HB_Z \prod_{i=1}^{H} B_i)}}{\sqrt{m}}$$

(35)

# B  SUPPLEMENTARY EXPERIMENTS

## B.1  EXPERIMENTS ON OTHER DATASETS

### B.1.1  EXPERIMENTS ON MNIST

We train the GANs using duality gap corresponding to WGAN-GP on MNIST. And compare our method with the traditional methods WGAN-GP. Specifically, we adopt a 3-layers CNN as the generator and a 3-layer CNN as the discriminator.In the subsection, $\lambda$ is 10. Number of discriminator iterators per generator iterators is 5. We take 100K iterations in all the experiments on MNIST datasets.

Figure 4 shows the Wasserstein Distance on MNIST datasets and Figure 5 shows the image generated after 100K iterations by the generator on MNIST datasets.

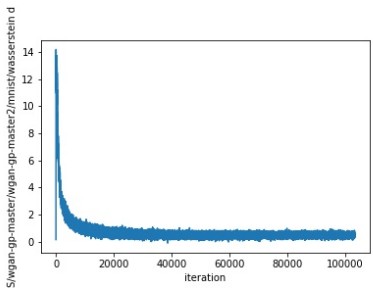

Figure 4: Wasserstein Distance on MNIST

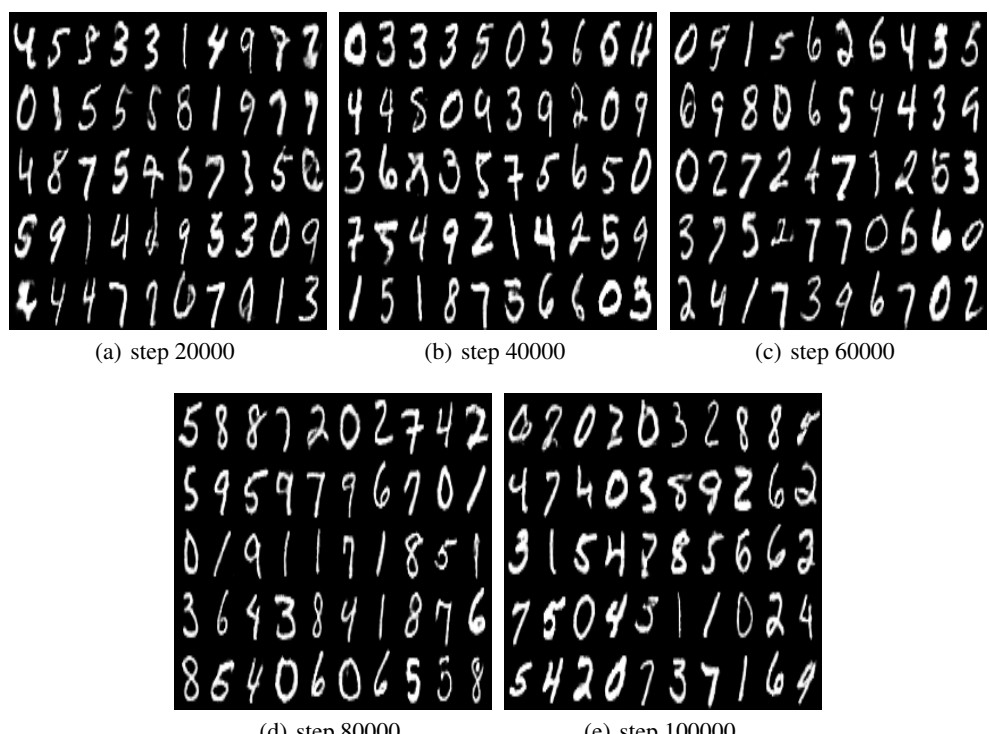

| (a) step 20000 | (b) step 40000 | (c) step 60000 |

| (d) step 80000 | (e) step 100000 |

Figure 5: result on MNIST datasets

### B.1.2 EXPERIMENTS ON TOY DATASETS

We train the the GANs using duality gap corresponding to WGAN-GP on three toy datasets with increasing difficulty: (1) RING: a mixture of 8 Gaussians, (2) GRID: a mixture of 25 Gaussians, (3)Swissroll. And compare our method with the traditional methods WGAN. Specifically, we adopt a 4-layers ReLU- with 512 hidden units as the generator and a 4-layer ReLU- with 512 hidden units as the discriminator. In the subsection, $\lambda$ is 0.1. Number of discriminator iterators per generator iterators is 5. We take 100K iterations in all the experiments on RING, 200K iterations on GRID and 200K iterations on Swissroll.

Figure 6 shows the Wasserstein Distance on the above three toy datasets and Figure 7 shows the image generated by the generator on the above three toy datasets. In the three figures, the yellow points represents the true data and the green points represent the generated data.

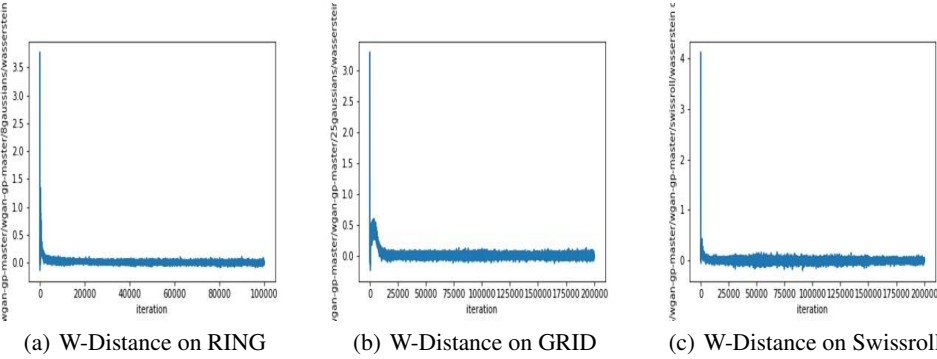

(a) W-Distance on RING    (b) W-Distance on GRID    (c) W-Distance on Swissroll

Figure 6: W-Distance on toy datasets

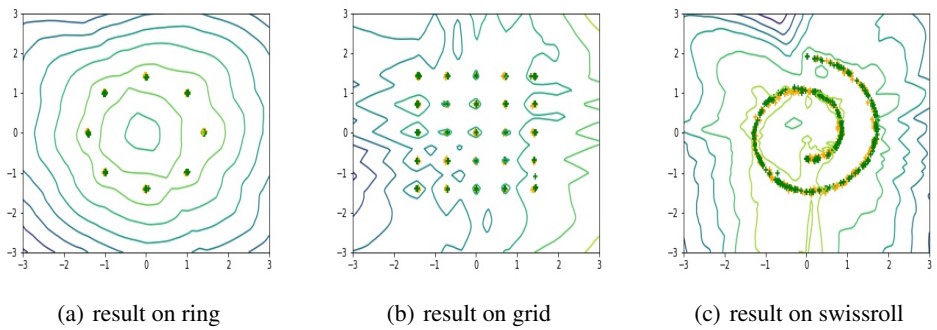

(a) result on ring    (b) result on grid    (c) result on swissroll

Figure 7: result on toy datasets

## B.2 TRAINING GANS USING DG CORRESPONDING TO OTHER GANS

Because for every traditional GANs which train GANs by minimizing $\mathcal{F}$-distance, we can find a duality gap corresponding to it. Thus, except the experiments in section 5, where the loss function is the duality gap corresponding to WGAN-GP, we also take WGAN, in consideration. For WGAN, we adopt a 4-layers CNN as the generator and a 3-layer CNN as the discriminator and the dataset is CIFAR-10.

We take 10K iterations in the experiments on CIFAR-10 and compare their inception scores and generated models. Figure 8 shows the inception score of WGAN and our method and Figure 9 shows their generated models:

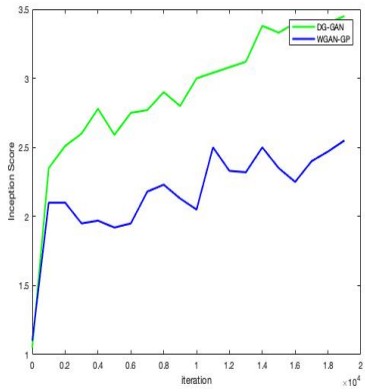

Figure 8: Inception score on CIFAR-10

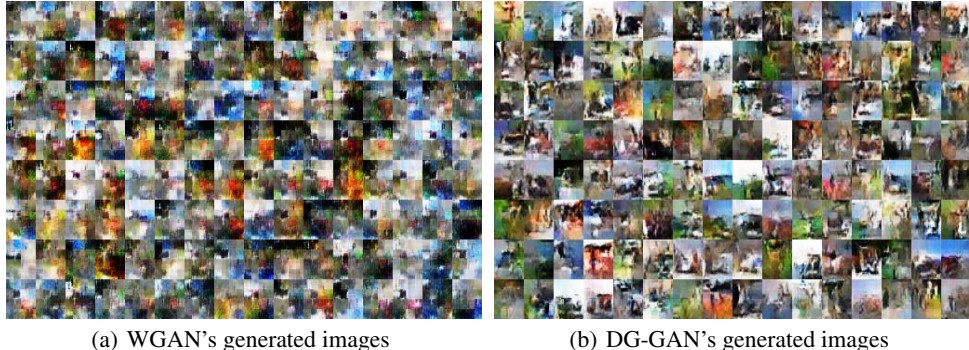

(a) WGAN's generated images      (b) DG-GAN's generated images

Figure 9: results on CIFAR-10

