# OpenReview forum: "DG-GAN: the GAN with the duality gap"
_ICLR.cc/2020/Conference — Reject_

### Official Review · AnonReviewer3 · 2019-10-07
**Official Blind Review #3**

**Rating:** 1

**Review:**

This paper has problems with clarity/polish and experimental design that are sufficiently severe
to merit rejection by themselves.

Regarding clarity/polish:
I am generally not super picky about these things, but there does have to be some standard.
This paper looks very hastily put together, especially pages 7 and 8.
There are many typos and unclear statements.
Just a few examples:

> Generative Adversarial Networks (GANs) are powerful framework for (in the abstract)
> be a good metric to evolution the difference (in the abstract)
> In the past few years, Generative Adversarial Networks (GANs) (Goodfellow et al., 2014) are impactful because it has shown lots of great results for many AI tasks, (first sentence)

> It means that there is no an unanimous metric to represent the difference between the true data distribution and the generated distribution
What does this mean? People have mostly settled on using FID for this.

> It is also difficult to know whether the generated distribution is close to the true distribution, and this is often observed by human eyes.
Isn't this just restating the point made in the first sentence?
Regardless, nobody really uses human evaluation anymore - so this is just not correct.

>  It means that if the original generator and discriminator are random, it is difficult to confirm that the generator and discriminator can converge to the ideal conclusion by training with given data.
But this paper doesn't propose a way to solve that problem, so it's strange to mention this here in this way.


These issues would maybe be excusable if not for the totally inadequate experimental validation.
A non-exhaustive list of methodological problems with the (single) experiment:

1. The experiment uses a single run each of the baseline and DG-GAN, when it's well-known that GAN training runs
have inter-run variance larger than the difference in score reported in Fig 1 and 2.

2. The models have not been trained for long enough.

3. The architecture of the neural networks used for the Generator and Discriminator is very non-standard, which
probably leads to:

4. The scores achieved by the baseline are very far from state of the art, making the comparison mostly useless,
and rendering the third claim from the introduction ("We propose an new algorithm with the new metric which demonstrates better results than state-of-the-art algorithms.") completely untrue.

In light of these other issues, I haven't checked the proofs.


**Experience Assessment:**

I have published in this field for several years.

**Review Assessment: Checking Correctness Of Derivations And Theory:**

I did not assess the derivations or theory.

**Review Assessment: Checking Correctness Of Experiments:**

I assessed the sensibility of the experiments.

**Review Assessment: Thoroughness In Paper Reading:**

I read the paper at least twice and used my best judgement in assessing the paper.

---

> ### Author Response · Authors · 2019-11-14
> **Response to Review#3**
>
> Thanks for your attention to our work.
> 1) For the presentation, we apologize for our typos and unclear statement in the paper. And your advice is so helpful. We will modify it.
>
> 2) For the experiment, we will train our experiments longer and modify our network. Thanks for your advice.

---

### Official Review · AnonReviewer1 · 2019-10-22
**Official Blind Review #1**

**Rating:** 1

**Review:**

I vote to reject the paper at this stage, mainly because of the following three points:

1) The motivation is unclear and overall structure of the paper is confusing. It should be better motivated why one should use the duality gap as an upper bound for the "F-distance". Minimizing the F-distance as is usually done seems like the more direct and simple approach. Since the results are far from state of the art, a clean and neat presentation of the theoretical advantages and contributions is crucial.

2) The presentation is not professional, hard to follow and the submission overall looks very rushed:
- In equations, please use \inf, \sup, and \text{...} for text such as distance, data, ...
- I have trouble understanding the overall idea behind Algorithm 1 and Eq. (22). What is the definition of f^* and g^* in Eq. (22)? Some explanatory text would be valuable.
- The set F in Definition 3.5 looks odd, as it appears to be recursive and might not be unique.
- The writing looks very rushed, and should be improved. For example, I have trouble understanding the sentence "So the existed algorithms should be heuristic or it can get a bad result even we train the neural networks with lots of datasets." in the introduction.
- The aspect ratio in Fig. 5 should be fixed.

3) The experiments are completely preliminary and not reasonable:
- The WGAN-GP baseline is very weak, i.e. does not show any reasonable generated images (Fig. 9). There are countless open pytorch implementations on GitHub which out-of-the-box produce much better results.
- The shown inception scores are far from state-of-the-art. It is unclear, why one should use the proposed duality gap GAN.

**Experience Assessment:**

I have published one or two papers in this area.

**Review Assessment: Checking Correctness Of Derivations And Theory:**

I did not assess the derivations or theory.

**Review Assessment: Checking Correctness Of Experiments:**

I assessed the sensibility of the experiments.

**Review Assessment: Thoroughness In Paper Reading:**

I read the paper at least twice and used my best judgement in assessing the paper.

---

> ### Author Response · Authors · 2019-11-14
> **Response to Review#1**
>
> Thanks for your attention to our work.
> 1) For the motivation, Because the traditional algorithms deal with GANs via a Markov chain:
> $(f_0,g_0)\rightarrow (f_1, g_0)\rightarrow (f_1,g_1) \rightarrow \cdots\rightarrow (f_{n},g_{n-1})\rightarrow (f_n,g_n)$. It is like a kind of reinforcement learning--- but the environment (Here it is $f$) is changing. And we want to view it from the angle of game theory. And then we try to minimize the new loss.
>
> 2) For the presentation: we will try to modify it. And we apologize that there are some typos about the $f^*$ and $g^*$ in the Eq.(22). The $f^*$ and $g^*$ means the dependent variable of duality gap. And the $\Leftrightarrow$ definition of $\mathcal{F}$ means equivalence. It can also be written as $f\in \mathcal{F}\rightarrow 1-f\in mathcal{F}$. And we will modify other improper presentation.
>
> 3)For the experiment: we will spend some time to train GANs with more iteration and modify it. Thanks

---

### Official Review · AnonReviewer2 · 2019-10-26
**Official Blind Review #2**

**Rating:** 1

**Review:**

This paper proposed to use the duality gap sup_f V(f, g*) – inf_g V(f*, g) as a metric for GAN training. It proves that this metric is an upper bound of F-distance. It also proves a generalization bound for this metric. Simulation resultson MNIST, CIFAR10, etc. are reported.

  The contribution of this paper is incremental due to the following reasons.

 1) The duality gap is only an upper bound of the F-distance. This means that if the duality gap is zero then the learned distribution is the true distribution. However, the converse is not necessarily true: even if the algorithm starts with the true distribution, the duality gap may not be zero. Thus the metric is not a proper metric.
  The proof of the upper bound is straightforward.

  2) Another issue is the gap between the min-max formulation and the real training algorithm. As for GAN, due to the inexact update, it is not really solving the min-max problem. For the proposed metric, it is also impossible to solve sup_f V(f, g*) and inf_g V(f*, g) to reasonable accuracy. Thus what the algorithm is really doing, perhaps, is to optimizing a new loss which is the sum of the original loss and and an extra term. Viewing it as a “duality gap” seems to be far from the practical training. This discrepancy exists for GANs, but it is a bigger issue for the duality gap interpretation.

  3) The simulation is not convincing. The reported FID for CIFAR10 using WGAN-GP is 54.4, which seems to be a bit high. I’m not sure whether it is due to parameter choice or due to weak D/G networks used in the simulation. If the paper cannot compare various architecture, it is more convincing to at least use some standard architecture, like DCGAN. Or at least report the parameter tuning effort made for getting the results.

**Experience Assessment:**

I have published one or two papers in this area.

**Review Assessment: Checking Correctness Of Derivations And Theory:**

I assessed the sensibility of the derivations and theory.

**Review Assessment: Checking Correctness Of Experiments:**

I assessed the sensibility of the experiments.

**Review Assessment: Thoroughness In Paper Reading:**

I read the paper at least twice and used my best judgement in assessing the paper.

---

> ### Author Response · Authors · 2019-11-14
> **Response to Review#2**
>
> Thanks for your attention to our work.
> 1) For the first problem that the duality gap is only an upper bound of F-distance. Our logic is that: a) There exists a condition s.t. duality gap = 0. b) If duality gap = 0, then the generator is the best one that can generate the true distribution. May be in the algorithm, we will miss the best generator because we do not get the equilibrium.
>
> 2) Our method may encounter the same problem as the traditional algorithm. It is a kind of Markov chain to train the Loss. And the essence of the algorithm is in fact to solve $\sup_f \inf_{g^*} V(f, g^*)$ and $\inf_g \sup_{f^*}V(f^*, g)$. We should consider some better algorithm to solve it.
>
> 3) For the experiments, we will do some modification and improve our network.

---

### Decision · Program_Chairs · 2019-12-19

**Decision:**

Reject

**Comment:**

This paper proposes looking at the duality gap to measure performance. However, the metric is just an upperbound on the true metric of interest, and therefore its value can be ambiguous.

The reviewers found the paper to be in an unacceptable form and was clearly hastily prepared. They were also skeptical about the novelty of the result as well as the comprehensiveness of the experiments.

This paper would require extensive revisions before any potential acceptance. Reject